# Sex Differences in Individuals at High Risk of Atrial Fibrillation: A Primary Care Community Cohort Study, 2015–2024

**DOI:** 10.3390/biomedicines13112814

**Published:** 2025-11-18

**Authors:** Jose Luis Clúa-Espuny, Anna Panisello-Tafalla, Alba Hernández-Pinilla, Josep Clua-Queralt, Eulàlia Múria-Subirats, Jorgina Lucas-Noll, Pedro Moltó-Balado, Teresa Forcadell-Arenas, Silvia Reverté-Villarroya

**Affiliations:** 1Equip d’Atenció Primària Tortosa Est, Gerència d’Atenció Primària i a la Comunitat de les Terres de l’Ebre, Institut Català de la Salut, 43500 Tortosa, Spain; jlclua.ebre.ics@gencat.cat (J.L.C.-E.); jcluaqueralt.ebre.ics@gencat.cat (J.C.-Q.); 2Fundació Institut Universitari per a la Recerca a l’Atenció Primària Jordi Gol i Gurina (IDIAPJGol), 43500 Tortosa, Spain; 3Servicio de Atención Primaria Camp de Tarragona, Institut Català de la Salut, 43761 Tarragona, Spain; ahernandez.tgn.ics@gencat.cat; 4Equip d’Atenció Primària Amposta, Gerència d’Atenció Primària i a la Comunitat de les Terres de l’Ebre, Institut Català de la Salut, 43500 Tortosa, Spain; emuria.ebre.ics@gencat.cat; 5Hospital Comarcal Mora d’Ebre, Salut Terres de l’Ebre, 43740 Mora d’Ebre, Spain; jlucas.hcme.ste@gencat.cat; 6Equip d’Atenció Primaria CSI Llíria, Departament de Salut de Arnau de Vilanova, Conselleria de Sanitat, 46160 Llíria, Spain; pemolto@gmail.com; 7Equip d’Atenció Primària Tortosa Oest, Gerència d’Atenció Primària i a la Comunitat de les Terres de l’Ebre, Institut Català de la Salut, 43500 Tortosa, Spain; tforcadella.ebre.ics@gencat.cat; 8Advanced Nursing Research Group, Nursing Department, Rovira i Virgili University, Biomedicine Doctoral Programme Campus Terres de l’Ebre, Av. De Remolins, 13, 43500 Tortosa, Spain

**Keywords:** atrial fibrillation, sex, aging, epidemiology, comorbidity, stroke, dementia, cognitive impairment, heart failure, ischemic disease, chronic kidney disease, primary care, risk stratification

## Abstract

**Background**: Sex differences in epidemiology and outcomes in atrial fibrillation (AF) are well documented, but their role in early detection and risk stratification in primary care remains unclear. **Methods**: This study used an observational, retrospective cohort design, including 9677 individuals identified as being at high risk (Quartile 4 of a validated prediction model) for developing atrial fibrillation, aged 65–95 years, and without prior AF or stroke in the Terres de l’Ebre health region (Catalonia, Spain). Incident AF and comorbidities prevalence were assessed from 1 January 2015 to 31 December 2024. Analyses compared sex-specific differences. **Results**: During follow-up, 3370 individuals (8.4%) developed AF, with higher incidence in men than women (9.9% vs. 7.0%, *p* < 0.001). In the high-risk subgroup (*n* = 9677), women had higher CHA_2_DS_2_-VA scores (4.10 vs. 3.84, *p* < 0.001) and greater prevalence of cognitive impairment (21.5% vs. 14.6%), while men more often presented with diabetes, ischemic cardiomyopathy, and peripheral vascular disease. Among new AF cases in this subgroup, men exhibited clustering of cardiometabolic conditions, whereas women showed higher cognitive decline. **Conclusions**: Distinct sex-specific patterns in comorbidity clustering and AF incidence were observed. These findings highlight the need for sex-tailored strategies for early AF detection and integrated risk management in primary care.

## 1. Introduction

Atrial fibrillation (AF) is one of the most commonly encountered heart conditions, with a broad impact on all health services across primary, secondary and tertiary care. The prevalence of AF is expected to double by 2050 because of the aging population, an increasing burden of comorbidities, improved awareness, and new technologies for detection [1]. The lifetime risk of AF is estimated to be 1 in 3–5 individuals over age 55 (37%; 95% CI 34.7–39.6%), and AF-related strokes are expected to rise by 34% in the coming decades. Current European guidelines highlight the need for early detection and structured, integrated management of AF [1,2,3].

AF is linked to multiple risk factors that contribute both to incident disease and increased risks of stroke, heart failure, cognitive decline, hospitalization, and all-cause mortality [4,5]. Sex differences have been consistently observed in AF, affecting pathogenesis, prevalence, clinical presentation, and outcomes [6,7]. Notably, women have a lower overall prevalence of AF but a higher risk of stroke [8,9]. They also tend to present at older ages, report more severe symptoms, and experience poorer quality of life and higher complication rates. Despite these findings, the implications of sex-specific differences for early detection, comorbidity clustering, and risk stratification remain insufficiently explored.

The Atrial Fibrillation Better Care (ABC) pathway [10] has demonstrated that comprehensive management—encompassing stroke prevention, symptom control, and cardiovascular risk optimization—reduces mortality and adverse events. Building on this, the 2024 European Society of Cardiology (ESC) guidelines propose the AF-CARE pathway (Comorbidity and risk factor management, Avoidance of stroke, Rate and rhythm control, and Evaluation and reassessment) as the central framework for AF management [1]. This patient-centered, multidisciplinary model emphasizes addressing underlying conditions and lifestyle factors, preventing thromboembolic events, optimizing rate and rhythm control, and ensuring continuous reassessment. However, most healthcare systems still adopt a reactive approach, with cardiology referral typically occurring only after disease onset or major complications such as stroke or heart failure. In contrast, Primary care is particularly well-positioned to implement a proactive and integrated approach through timely identification and intervention [11] on high-risk individuals before a serious problem arises.

Although several AF prediction scores have been developed [12,13,14,15,16] from primary care electronic health records, their discriminative performance across different clinical settings is unknown and they are not widely integrated into practice. Screening a high-risk population has improved detection and may provide an opportunity to address co-morbid conditions and prevent future cardiovascular outcomes like stroke. Optimal strategies and settings for atrial fibrillation screening remain undefined. A major gap is the lack of sex-specific risk assessment tools, despite well-documented differences between women and men. Personalized risk prediction for AF incidence, AF progression, and associated outcomes remains challenging. The aim of this study was to analyze sex differences in AF incidence and comorbidities in a large community-based cohort, with a focus on high-risk population.

## 2. Materials and Methods

### 2.1. Study Design and Setting

The study employed an observational, retrospective cohort design, including individuals identified as being at high risk (Quartile 4) for developing AF. This study analyzed data collected from 1 January 2015 to 31 December 2024. The research was undertaken within the framework of the project Gender Perspective on Cardiovascular Diseases in the Terres de l’Ebre (GECA-TE), part of a doctoral research program. The present work represents a substudy of patients included in previously published investigations [11,17,18,19], with the objective of identifying sex differences in the epidemiology of AF and cardiovascular health. Specifically, the study examined differences in epidemiological patterns, clinical presentation, risk factors, and outcomes within the geographical setting of the Terres de l’Ebre region (Appendix A). The study report followed the Strengthening the Reporting of Observational Studies in 104 Epidemiology (STROBE) guidelines. The research protocol was reviewed and approved by the Ethics Committee of the Jordi Gol University Institute of Primary Care Research (registration number 24/187-P; approval date 12 February 2025).

### 2.2. Study Scope

The study was carried out in the Terres de l’Ebre Health Region, located in southern Catalonia, Spain (Appendix A). The region includes 178,112 inhabitants (49.6% women) distributed across 54 municipalities, with an average population density of 53.8 inhabitants/km^2^, compared with 241.8 inhabitants/km^2^ in Catalonia overall [20,21]. The population is characterized by advanced aging, with an aging index of 159.5, higher than that of Catalonia (131.3) and Spain (118.4) [22]. This index was calculated as the ratio between individuals aged ≥ 65 years and those aged < 15 years per 100 inhabitants. The population aged ≥ 65 years represented 31.1% of the total. The average per capita income was lower than that of Catalonia (77.4% vs. 100%) [23].

The territory comprises four counties and 11 primary care teams (EAPs), all managed by the Catalan Health Institute (ICS) under the authority of the Department of Health (CatSalut). Specialized care is provided at the reference hospital, Hospital Verge de la Cinta (Tortosa), also publicly managed by the ICS. The EAPs operate as independent clinical-functional units. Nearly all residents (99.2%) have an active digital health record in the Shared Health Record of Catalonia (HCC3), enabling continuous monitoring across all levels of care.

### 2.3. Data Collection and Information Sources

Data were extracted by the Information and Communication Technology Department from the CMBD (Minimum Basic Data Set), a clinical-administrative database that compiles detailed information on healthcare episodes, particularly demographic and clinical data, as well as information on patient procedures, such as diagnoses, treatments, length of stay, and mortality for use in health planning, evaluation, and research; and the Shared Health Record of Catalonia (HCC3), which integrates information from primary and specialist care, hospital admissions, prescriptions, and mortality records. Collected variables included demographic data, comorbidities (hypertension, diabetes, ischemic heart disease, peripheral vascular disease, chronic kidney disease, and cognitive impairment), AF diagnosis, and mortality. Risk scores, including the CHA_2_DS_2_-VA (as recommended in the 2024 ESC/EACTS Guidelines on AF) and the Charlson Comorbidity Index, as well as healthcare resource utilization. Diagnoses were coded according to the International Classification of Diseases, 10th Revision (ICD-10). All data were processed in encrypted format.

### 2.4. Study Population

The study population included 40,079 individuals without a prior diagnosis of AF (Figure 1, flowchart). The primary outcomes were sex-specific incidence of AF and cardiovascular comorbidities, cognitive decline, and all-cause mortality among individuals classified as high risk for AF. The study hypothesis was that incidence of these outcomes prior to AF diagnosis would not differ significantly between men and women.

### 2.5. Inclusion and Exclusion Criteria

Inclusion criteria: Patients aged 65–95 years were included if they were in the highest quartile (Q4th) of AF risk according to a validated prediction model [12,24], had active medical records accessible through the HCC3/CMBD systems, had no prior diagnosis of AF, and were residents assigned to one of the region’s EAPs. The risk score used to stratify patients was developed and internally validated in a previous population-based cohort [12,24]. The model incorporates age, sex, hypertension, diabetes, vascular disease, heart failure, and body mass index, with predictive performance AUC = 0.78 (95% CI 0.75–0.81). The present study enrolled individuals in the top quartile (Q4) of this score, corresponding to those with ≥75th percentile of predicted AF risk.

Exclusion criteria: Patients with prior stroke to avoid reverse causality between previous cerebrovascular events and subsequent AF detection. Individuals with pacemakers or defibrillators to minimize detection bias associated with continuous rhythm monitoring. In addition, patients who lacked an AF risk index [12,24] or who resided outside the Terres de l’Ebre region were excluded.

### 2.6. Variables

Clinical events and comorbidities were systematically collected from primary care practices and supplemented through linkage with AI-driven electronic health databases. Follow-up extended until death, loss to follow-up, or 31 December 2024, whichever occurred first. AF diagnosis followed the European Society of Cardiology (ESC) guidelines. Upon AF diagnosis, the event was timestamped, and clinical data for all participants—both those who developed AF and those who did not—were extracted at the end of follow-up. This ensured consistency in data collection across groups.

Cardiovascular risk factors and comorbidities were identified using ICD-10 codes: cerebrovascular disease (I63, G45), heart failure (I50–I51), ischemic heart disease (I20–I25), hypertension (I10–I15), hypercholesterolemia (E78), diabetes mellitus (E10–E14), body mass index (BMI), chronic kidney disease (N18), and estimated glomerular filtration rate (eGFR, mL/min/1.73 m^2^). Only incident events were included, verified as occurring after the AF diagnosis. Clinical assessments included the Charlson Comorbidity Index, CHA_2_DS_2_-VA score, and Pfeiffer Short Mental Status Questionnaire. Biological sex, as registered in patient databases, was used for analysis. Vital status was recorded at the end of follow-up.

### 2.7. Statistical Analysis

Population characteristics were described using descriptive statistics. Continuous variables are presented as mean ± standard deviation if normally distributed, or median values otherwise; categorical variables are reported as counts and percentages. Student’s *t*-test was applied for continuous variables, and chi-square tests for categorical variables. Incidence rates were calculated as events per 1000 person-years of follow-up. Person-time was computed as the cumulative follow-up time from AF diagnosis until study end or censoring. Incidence rate ratios (IRRs) were calculated to compare incidence between groups, as these account for time at risk. Two-sided *p*-values < 0.05 were considered statistically significant. All analyses were performed using IBM SPSS Statistics, version 21.0.

## 3. Results

### 3.1. Overall Cohort

Table 1 provides a summary of the baseline characteristics of the study cohort, which consisted of 40,079 individuals between the ages of 65 and 95 years. Data are stratified by sex for the total population. During the study period, 3370 new cases of AF were diagnosed, resulting in an overall prevalence of 8.4%. A significant sex-based difference was noted, with a higher prevalence in men (9.9%) compared to women (7.0%) (*p* < 0.001). The prevalence of AF in the territory is showed in Appendix A. The average age was slightly higher for women (77.6 ± 6.63 years) than for men (77.28 ± 6.56 years), a difference that was also statistically significant (*p* < 0.001). Significant sex-related differences were observed across several comorbidities. Men had a higher prevalence of diabetes mellitus (29.5% vs. 21.9%), peripheral vascular disease (10.2% vs. 3.9%), and ischemic cardiomyopathy (10.6% vs. 4.5%). Conversely, women exhibited a higher prevalence of dyslipidemia (51.7% vs. 43.0%) and dementia or cognitive impairment (10.8% vs. 7.4%). These differences highlight the distinct comorbidity profiles between sexes within this older cohort.

### 3.2. High-Risk for Atrial Fibrillation Subgroup (Quartile 4)

Table 2 presents the characteristics of the high-risk subgroup, defined as individuals in the fourth quartile of AF risk (*n* = 9677). There is a higher prevalence of classic cardiovascular risk factors associated with a heightened AF risk in both sexes and showed sex-specific differences. Within this subgroup, men exhibited a higher mean CHA_2_DS_2_-VA score compared with women (4.10 ± 0.97 vs. 3.84 ± 0.88, *p* < 0.001). Men also demonstrated higher prevalence of new-onset AF (20.4% vs. 15.6%), heart failure (29.2% vs. 26.0%), diabetes mellitus (53.4% vs. 44.3%), stroke (10.8% vs. 9.1%), vascular peripheral disease (24.2% vs. 11.4%), and ischemic cardiomyopathy (26.0% vs. 13.1%). Conversely, women in the high-risk cohort had a higher prevalence of dementia or cognitive impairment (21.5% vs. 14.6%) and dyslipidemia (57.0% vs. 49.1%).

### 3.3. New-Onset AF in the High-Risk AF (Quartile 4) Subgroup

Table 3 presents a sex-stratified comparison of clinical characteristics for individuals in the highest risk quartile based on the incidence of new-onset AF. The data reveals that **20.4%** of men and **15.62%** of women developed AF during the follow-up period. 563 individuals were excluded from the analysis due to a lack of available data. The table shows significant differences between men who developed AF and those who did not, with those developing AF having a higher CHA_2_DS_2_-VA score (4.43 vs. 4.04), higher prevalence of heart failure, stroke, and chronic kidney disease. Among women, those who developed AF exhibited higher CHA_2_DS_2_-VA score (4.18 ± 0.9 vs. 3.80 ± 0.8), higher prevalence of heart failure, stroke, vascular peripheral disease, ischemic cardiomyopathy, and chronic kidney disease. They also showed higher Charlson comorbidity index values (2.30 ± 1.4 vs. 1.90 ± 1.38), more hospital visits (0.65 ± 1.5 vs. 0.38 ± 1.2), and greater polypharmacy (8.84 ± 5.2 vs. 7.54 ± 4.9).

Comparing sex-specific differences within the Q4th-AF subgroup, men exhibited higher CHA_2_DS_2_-VA and Charlson comorbidity index scores, along with a greater prevalence of diabetes mellitus, peripheral vascular disease, ischemic cardiomyopathy, and obstructive sleep apnea. In contrast, women presented with higher body mass index and Pfeiffer scores, as well as a greater prevalence of dementia and cognitive impairment. Among all individuals with new-onset AF (*n* = 863, 20.4%), the prevalence of several comorbidities was higher in the new-onset AF group, including heart failure (55.2% vs. 22.5%), ischemic cardiomyopathy (27.2% vs. 25.9%), peripheral vascular disease (25.0% vs. 23.9%), and chronic kidney disease (41.6% vs. 33.1%). Furthermore, characteristics were significantly more prevalent compared to those without AF: higher CHA_2_DS_2_-VA scores (4.43 ± 0.9 vs. 4.04 ± 0.9), higher Charlson comorbidity index scores (2.83 ± 1.4 vs. 2.48 ± 1.5), and a higher number of active medications (8.8 ± 4.8 vs. 6.76 ± 4.8).

In summary, new-onset AF was associated with a substantial comorbidity burden in both sexes, with men exhibiting a predominance of cardiometabolic risk factors and vascular disease, and women more frequently presenting with cognitive impairment. The percentage of individuals with heart failure, vascular peripheral disease and ischemic cardiomyopathy was significantly higher in both men and women who developed new AF compared to their counterparts who did not.

### 3.4. Clinical Outcomes

Table 4 shows the incidence of cardiovascular events and rate ratios by sex. The data are stratified by AF status within the Q4th, with event rates expressed per 1000 person-years. Among individuals with a new AF diagnosis, men had a higher incidence rate for ischemic heart disease and peripheral artery disease. Women with new-onset AF, however, had a higher rate of chronic kidney disease. No significant differences were detected between the sexes for stroke, heart failure, cognitive impairment, or all-cause mortality in this group. Conversely, among individuals without AF, women had a higher incidence rate of both cognitive impairment and mortality compared to men.

## 4. Discussion

This study focuses on individuals at high risk of developing AF, a population in which cardiovascular comorbidities are frequent, often interrelated and challenging to distinguish [3,25]. Consistent with prior evidence, men developed AF earlier and more frequently, while women presented later with a higher comorbidity burden [26,27]. However, evidence is lacking regarding potential sex-related differences among individuals at high risk of developing AF. This study aimed to characterize sex-specific clinical patterns within individuals already identified as high-risk for AF, rather than to identify new predictors. The analysis explores heterogeneity within the upper quartile of a validated prediction model, complementing prior work on risk development. Addressing this gap could refine risk stratification, screening, and preventive strategies. Appendix A provides a summary of sex differences in patients at high risk of AF (quartile 4) and new AF. These sex-related differences should not be interpreted as causal or as direct consequences of AF, but rather as two complementary patterns:

Intra-sex differences: Comorbidities are largely comparable across sexes, although men exhibit higher mortality rates after diagnosis AF and women in Quartile 4. Within each sex, individuals with AF may differ considerably from those without AF, primarily due to variations in comorbidity profiles and their associated outcomes. Many of these factors are already incorporated into clinical prediction scores [12,13,14,15,16]. Reinforcing their systematic use in primary care—together with comprehensive prevention strategies, opportunistic screening initiatives, and targeted management of comorbidities—could represent a pragmatic approach to mitigating AF risk and improving long-term outcomes [28].

Inter-sex differences: Men and women differ in their comorbidity profiles. Beyond these differences, sex itself influences AF incidence, clinical presentation, and prognosis. Biological mechanisms, including hormonal regulation, atrial remodeling, and disparities in healthcare utilization, may interact with social determinants such as health-seeking behavior and unequal access to treatment [29]. These interactions may partly account for the observed disparities in anticoagulation, rhythm control, and clinical outcomes. Future research directed toward the development and validation of sex-specific risk scores could enhance the understanding of these mechanisms and support follow-up strategies tailored to sex-specific risks [30].

Within high-risk individuals, intra-sex comparisons revealed that AF cases carried a greater comorbidity load than non-AF counterparts, reinforcing the value of systematic use of prediction scores in primary care alongside prevention and opportunistic screening. Inter-sex comparisons showed distinct comorbidity patterns: women more often had hypertension, cognitive impairment, dyslipidemia, and mortality, whereas men more often had diabetes, vascular disease, ischemic cardiomyopathy, OSAHS, and higher Charlson index. Incident AF in both sexes was associated with heart failure, stroke, chronic kidney disease, and OSAHS. Excess mortality was more pronounced in men.

Age remained a major determinant, with AF incidence rising linearly [31,32]. In the very elderly, AF is frequently asymptomatic or manifests with nonspecific symptoms, which may delay diagnosis and consequently heighten the risk of adverse outcomes and reduce survival [33]. Additionally, older patients may experience a lower benefit from rhythm control strategies, such as antiarrhythmic medications and ablation techniques. Findings from European studies showed the ablation procedures for atrial fibrillation were less frequently performed in women [34]. The presence of frailty and cognitive decline can further complicate management by affecting medication adherence and the patient’s capacity to engage in treatment decisions. These two conditions collectively underscore the importance of early AF diagnosis and the management of associated vascular comorbidities.

Within the 4th Quartile, women exhibited a higher prevalence of cognitive impairment and greater mortality compared to men. However, these differences were not statistically significant once an atrial fibrillation diagnosis was established. Differences in sex for prior cardiovascular disease prevalence are well established major risk factors for incident AF [35,36]. The cumulative risk of developing AF was higher in men than in women. The presence of microvascular disease, particularly in those with type 2 diabetes, was independently associated with higher risk of incident atrial fibrillation [37]. Attributable risk over time for atrial fibrillation associated with higher body mass index and most cardiovascular factors included in the CHA_2_DS_2_-VA score have been previously described and incorporated into AF risk scores [24,38,39]. The significant increase in heart failure prevalence following an atrial fibrillation diagnosis warrants attention, as the co-occurrence of these conditions is associated with a worse prognosis, particularly for women, who face increased mortality and greater symptom burden compared to men [40]. AF is a known risk factor for cognitive impairment and dementia, and changes like brain atrophy independent of clinical stroke. High-risk individuals, particularly women, also face an increased risk of these conditions, making screening for AF and cognitive impairment a crucial preventive measure for this population [41,42].

The overall age-adjusted mortality rates (AAMRs) for AF-related deaths among adults aged ≥ 65 years showed a steady increase between 1999 and 2020. This rise was accompanied by higher AAMRs among men for conditions frequently associated with an AF diagnosis, including coronary artery disease, heart failure, cognitive impairment, obstructive sleep apnea, and metabolic syndrome [43,44,45]. These conditions and their sex-specific prevalence highlight the substantial mortality burden associated with AF, particularly through its contribution to stroke, heart failure, and cognitive impairment [46]. Thus, AF poses a significant risk for premature mortality. The early identification of high-risk subgroups is critical for implementing targeted preventive strategies and optimizing care for individuals most likely to benefit.

Emerging evidence indicates that an intrinsically prothrombotic atrial substrate may precede the onset of atrial fibrillation and contribute to thromboembolic events independently of the arrhythmia itself. Notably, the incidence of Major Adverse Cardiovascular Events increases progressively from individuals at low risk of atrial fibrillation to those at high risk, with a significantly higher incidence in patients with established AF [19,28]. These patients also frequently present with multiple comorbidities that exert a substantial impact on prognosis and therefore require careful clinical attention. Reflecting this evolving understanding, updated guidelines now conceptualize AF as a continuum that necessitates stage-specific strategies, with particular emphasis on and the comprehensive management of comorbidities and anticoagulation [1,5]

Early detection of AF is paramount because it enables early initiation of treatment that can lower the risk of adverse cardiovascular outcomes, particularly in older patients > 75 or with a CHA_2_DS_2_-VA score ≥ 2 and cardiovascular conditions or those with existing comorbidities, as demonstrated by the EAST-AFNET 4 study [47]. Photoplethysmography devices have demonstrated significant utility in the screening and detection of new cases of AF among high-risk patient populations. Studies have shown promising results, with a new AF diagnosis rate of approximately 1 in 10–14 screened individuals, though this rate is contingent on the specific device and patient cohort [3,11,48]. The use of this screening methodology could potentially diagnose a substantial number of additional AF cases. Based on our analysis, the consistent application of these strategies could have led to the identification of up to 600 additional cases of atrial fibrillation, which would supplement the 1724 cases diagnosed during the study period. This finding highlights the substantial potential for proactive screening and early detection of AF diagnosis and underscores a critical need for evaluating and integrating risk prediction tools within primary care. A more targeted approach creates a more efficient care pathway. Once a high-risk patient is identified, they can be directly routed to confirmatory diagnostic tests, which reduces delays and improves the continuity of care [1,10].

The findings demonstrate sex-specific differences in cardiovascular profiles, which are associated with variations in the incidence of atrial fibrillation and are consistent with previous reports describing greater comorbidity burden and worse outcomes in women with AF. Nevertheless, several potentially relevant factors were not considered such as AF-related fibrotic remodeling [49], emerging sex-specific factors [50,51] social determinants of health [52], and AF burden [53]. There is limited evidence on how sex differences in atrial electrical remodeling directly contribute to the observed disparities in risk [7,54]. It remains unclear whether these sex differences are related to intrinsic biological disparities in atrial remodeling or by a greater burden of comorbidities in one sex versus the other.

The main strengths of this study include its community-based design, large sample size, and long follow-up period, which enhance the external validity of our findings and reflect real-world primary care practice. While most large-scale studies derive from hospital or registry data, our findings suggest that sex-specific differences are also evident in primary care populations and may influence preventive strategies. The longitudinal design (2015–2024) further strengthens the relevance of our results, allowing assessment of temporal changes in risk factor management. Limitations include its retrospective design, lack of detailed information on pharmacological treatment, socioeconomic status, AF burden, and fibrotic remodeling, which may influence AF risk. Although sex and comorbidities form part of the variables included in the risk score, our subgroup analysis examined residual heterogeneity within the highest-risk quartile and the persistence of sex-specific differences within this stratum suggests that factors beyond the model’s weighting—such as cognitive decline or differential healthcare use—may further modulate AF incidence. Eventually multivariable modeling could improve robustness, but we opted for univariate analyses, because the high-risk subgroup was defined using a composite score that already integrates the main confounding variables. Further adjustment would have introduced collinearity and model instability. Despite these limitations, the findings support the recommendations of the AF-CARE Guidelines, particularly regarding the importance of early detection and management of comorbidities, highlighting the need for tailored strategies to address sex-specific risk patterns.

These findings should be interpreted within the context of a cohort defined by an internally risk score and primarily serve to generate hypotheses for sex-specific preventive strategies rather than to infer causality or establish universal risk gradients. Given the rapidly increasing global burden of atrial fibrillation, our findings highlight the need for a targeted public health strategy, particularly within the primary care setting. Our study suggests three core objectives to modify the impact of this condition: Enhancing Awareness of Atrial Fibrillation, Implementing Proactive Screening Strategies, and Targeting personalized Modifiable Risk Factors.

## 5. Conclusions

Sex differences were evident within this large population-based cohort. Men exhibited higher comorbidity clustering, particularly cardiometabolic disorders, while women more often presented with cognitive decline and greater comorbidity burden at diagnosis. These results underline the importance of adopting a sex-sensitive approach to AF prevention, early detection, and integrated management within primary care. These results reinforce guideline recommendations for risk stratification, tailored strategies for early AF detection and comprehensive management of comorbidities, supporting the implementation of sex-specific strategies to optimize patient outcomes, and contribute to more equitable healthcare.

## Figures and Tables

**Figure 1 biomedicines-13-02814-f001:**
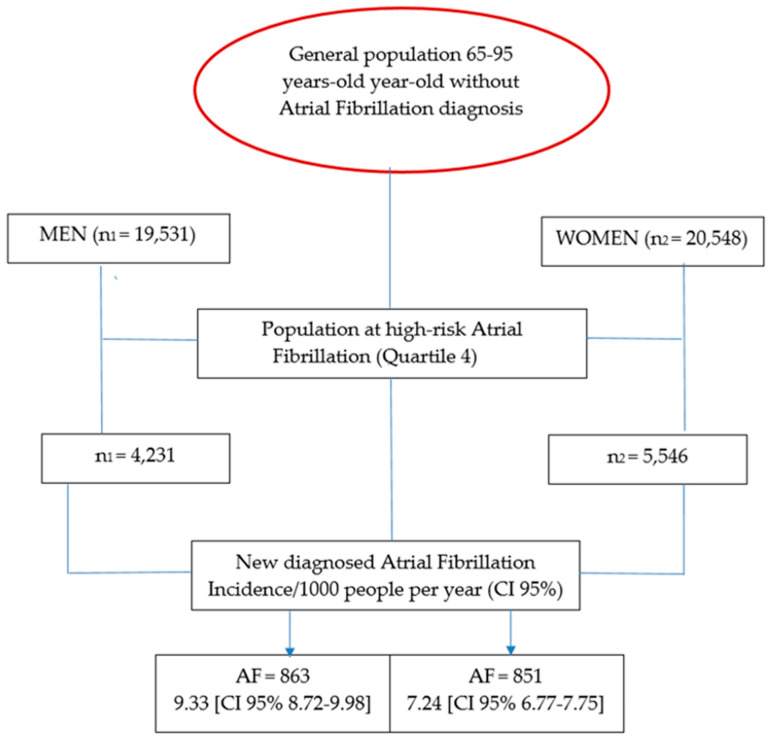
Flow-chart schedule [12,24].

**Table 1 biomedicines-13-02814-t001:** Baseline characteristics of the study population according to sex (*n* = 40,079).

Variables	Men	(%)	Women	(%)	*p*	All (%)
All (*n* %)	19,531	48.7%	20,548	51.3%	-	40,079
New AF	1928	9.9%	1442	7.0%	<0.001	3370 (8.4%)
Age average	77.28 ± 6.56		77.6 ± 6.63		<0.001	77.4 ± 6.60
CHA_2_DS_2_-VA	2.8 ± 1.1		2.6 ± 1.1		<0.001	2.68 ± 1.1
Heart failure	1790	9.2%	1558	7.6%	<0.001	3348 (8.4%)
Hypertension arterial	11,526	59.0%	12,218	59.5%	0.360	23,744 (59.2%)
Age 65 to 74 years	7748	39.6%	7765	37.8%	<0.001	15,513 (38.7%)
Age ≥ 75 years	11,783	60.3%	12,783	62.6%	<0.001	24,566 (61.3%)
Diabetes mellitus	5763	29.5%	4498	21.9%	<0.001	10,261 (25.6%)
Stroke/TIA/Systemic embolism	843	4.7%	724	3.5%	<0.001	1567 (3.9%)
Peripheral vascular disease	1983	10.2%	798	3.9%	<0.001	2781 (6.9%)
Ischemic heart disease	2073	10.6%	917	4.5%	<0.001	2990 (7.5%)
BMI ^1^ (kg/m^2^)	28.1 ± 4.5		28.4 ± 5.7		<0.001	28.3 ± 5.2
Charlson index	1.5 ± 1.4		1.2 ± 1.2		<0.001	1.38 ± 1.9
Dementia/cognitive impairment	1452	7.4%	2225	10.8%	<0.001	3677 (9.2%)
Pfeiffer score	2.72 ± 3.20		3.63 ± 3.35		<0.001	3.23 ± 3.3
Chronic Kidney Disease	3181	16.3%	3080	15.0%	<0.001	6261 (15.6%)
Glomerular filtration rate (mL/min/1.73 m^2^)	72.2 ± 18.0		73.4 ± 17.6		<0.001	72.9 ± 17.7
OSAHS ^2^	966	4.9%	473	2.3%	<0.001	1439 (3.6%)
Dyslipidaemia	8394	43.0%	10,623	51.7%	<0.001	19,017 (47.5%)
Statins	6006	30.8%	6326	30.8%	0.940	12,332 (30.8%)
Antiplatelet therapy	3411	17.5%	2335	11.4%	<0001	5746 (14.3%)
Anticoagulation	1977	10.1%	1468	7.1%	<0.001	3445 (8.6%)
Hospital visits	0.36 ± 1.3		0.27 ± 0.95		<0.001	0.31 ± 1.15
Active medications	5.17 ± 4.3		5.76 ± 4.40		<0.001	5.48 ± 4.36
Death all-causes	14,492	74.2%	17,173	83.6%	<0.001	31,665 (79.0%)

^1^ BMI: Body Mass index; ^2^ OSAHS: Obstructive Sleep Apnoea–Hypopnea Syndrome.

**Table 2 biomedicines-13-02814-t002:** Baseline characteristics by sex in the population at high risk of AF (quartile 4).

Variables	Men	(%)	Women	(%)	*p*	All (%)
All (*n* %)	4231	43.72%	5446	56.27%		9677
Age average	84.6 ± 6.9		84.7 ± 6.7		0.474	84.66 ± 6.76
65–74 years	131	3.1%	97	1.78%		228 (2.35%)
≥75 years	4100	(96.90%)	5349	98.21%		9449 (97.64%)
CHA_2_DS_2_-VA	4.10 ± 0.97		3.84 ± 0.88		<0.001	3.96 ± 0.9
New AF	863	20.4%	851	15.62%	<0.001	1714 (17.7%)
Heart failure	1236	29.2%	1415	26.0%	<0.001	2651 (27.4%)
Hypertension arterial	3708	87.6%	4916	90.3%	<0.001	8624 (89.1%)
Diabetes mellitus	2261	53.4%	2414	44.3%	<0.001	4675 (48.3%)
Stroke/TIA/Systemic embolism	458	10.8%	496	9.1%	0.005	954 (9.9%)
Peripheral vascular disease	1025	24.2%	619	11.4%	<0.001	1644 (17.0%)
Ischemic heart disease	1100	26.0%	714	13.1%	<0.001	1814 (18.7%)
BMI ^1^ (kg/m^2^)	30.2 ± 5.0		31.23 ± 5.9		<0.001	30.78 ± 5.5
Charlson index	2.54 ± 1.5		1.96 ± 1.3		<0.001	2.21 ± 1.4
Dementia/cognitive impairment	616	14.6%	1173	21.5%	<0.001	1789 (18.5%)
Pfeiffer score	2.87 ± 3.0		3.93 ± 3.1		<0.001	3.54 ± 3.1
Chronic Kidney Disease	1469	34.7%	1881	34.5%	0.863	3350 (34.6%)
Glomerular filtration rate (mL/min/1.73 m^2^)	60.55 ± 19.6		60.28 ± 19.6		0.623	60.4 ± 19.6
OSAHS ^2^	274	6.5%	160	2.9%	<0.001	434 (4.5%)
Dyslipidaemia	2079	49.1%	3104	57.0%	<0.001	5183 (53.6%)
Statins	1436	33.9%	1664	30.6%	<0.001	3100 (32.0%)
Antiplatelet therapy	1098	26.0%	1079	19.8%	<0.001	2177 (22.5%)
Anticoagulation	792	18.7%	772	14.2%	<0.001	1564 (16.2%)
VKAs ^3^	330	7.8%	318	5.8%	<0.001	648 (6.7%)
NOACs ^4^	463	10.9%	457	8.4%	<0.001	920 (9.5%)
Hospital visits	0.56 ± 1.51		0.40 ± 1.21		<0.001	0.48 ± 1.43
Active medications	7.07 ± 4.7		7.32 ± 4.6		0.009	7.38 ± 4.9
Death all-causes	2456	58.0%	3543	65.1%	<0.001	5999 (62.0%)

^1^ BMI: Body Mass index; ^2^ OSAHS: Obstructive Sleep Apnea–Hypopnea Syndrome; ^3^ VKAs: vitamin K antagonists; ^4^ NOACs: Non-vitamin K Antagonist Oral Anticoagulants.

**Table 3 biomedicines-13-02814-t003:** Comparative characteristics by sex of high-risk individuals (quartile 4) without AF versus with new-onset AF.

Variables	Men			Women		
	Q4th-no AF	AF	*p*	Q4th-no AF	AF	*p*
All (*n* %)	3151	863 (20.4%)		4249	851 (15.6%)	
Age average	85.50 ± 6.0	84.5 ± 5.8 **	<0.001	86.33 ± 5.6	86.3 ± 5.5	0.884
CHA_2_DS_2_-VA	4.04 ± 0.9	4.43 ± 0.9 **	<0.001	3.80 ± 0.8	4.18 ± 0.9	<0.001
Heart failure	708 (22.5%)	476 (55.2%)	<0.001	882 (20.8%)	442 (51.9%)	<0.001
Hypertension arterial	2771 (87.9%) **	778 (90.2%)	0.0823	3866 (91.0%)	764 (89.8%)	0.294
Diabetes mellitus	1725 (54.7%) **	457 (53.0%) **	0.3699	1937 (45.6%)	360 (42.3%)	0.085
Stroke/TIA/Systemic embolism	310 (9.8%) **	133 (15.4%)	<0.001	352 (8.3%)	125 (14.7%)	<0.001
Vascular peripheral disease	753 (23.9%) **	216 (25.0%) **	0.5199	470 (11.1%)	122 (14.3%)	0.007
Ischemic cardiomyopathy	816 (25.9%) **	235 (27.2%) **	0.4556	536 (12.6%)	141 (16.6%)	0.002
BMI ^1^ (kg/m^2^)	30.25 ± 4.9 **	30.46 ± 5.5 **	0.3095	31.45 ± 5.8	31.23 ± 6.1	0.379
Charlson index	2.48 ± 1.5 **	2.83 ± 1.4 **	<0.001	1.90 ± 1.38	2.30 ± 1.4	<0.001
Dementia/cognitive impairment	440 (14.0%) **	136 (15.8%)	0.2013	904 (21.3%)	154 (18.1%)	0.041
Pfeiffer score	2.94 ± 3.1 **	2.51 ± 2.7 **	<0.001	3.86 ± 3.2	3.48 ± 2.9	0.001
Chronic Kidney Disease	1042 (33.1%)	359 (41.6%)	<0.001	1406 (33.1%)	553 (39.4%)	<0.001
Glomerular Filtration rate (mL/min/1.73 m^2^)	61.33 ± 19.5	58.5 ± 19.7	<0.001	61.51 ± 19.3	56.6 ± 19.8	<0.001
OSAHS ^2^	190 (6.0%) **	82 (9.5%) **	<0.001	121 (2.8%)	38 (4.5%)	0.017
Dyslipidaemia	1551 (49.2%) **	454 (52.6%)	0.0848	2456 (57.8%)	474 (55.7%)	0.273
Statins	1086 (34.5%) **	326 (37.8%) **	0.0777	1353 (31.8%)	259 (30.4%)	0.443
Antiplatelet therapy	1015 (32.2%) **	55 (6.4%)	<0.001	993 (23.4%)	39 (4.6%)	<0.001
Hospital visits	0.52 ± 1.6 **	0.75 ± 1.7	<0.001	0.38 ± 1.2	0.65 ± 1.5	<0.001
Active medications	6.76 ± 4.8 **	8.8 ± 4.8	<0.001	7.54 ± 4.9	8.84 ± 5.2	<0.001
Death all-causes	1832 (58.1%) **	585 (67.8%)	<0.001	2865 (67.4%)	563 (66.2%)	0.496

** Significant statistical difference women vs. men. ^1^ BMI: Body Mass Index; ^2^ OSAHS: Obstructive Sleep Apnea–Hypopnea Syndrome.

**Table 4 biomedicines-13-02814-t004:** Sex-specific incidence of cardiovascular comorbidities and incidence rate ratios.

	Men		Women		Incidence Rate Ratios Men/Women	
Incidence/1000 People per Year (CI95%)	High AF-Risk (Q4th)	New AF	High AF-Risk (Q4th)	New AF	OR Q4th/Q4th (CI95%)	OR AF/AF (CI95%)
N	3151	863	4249	851		
AF Incidence/1000 people per year (CI95%)		9.33 (8.72–9.98)	-	7.24 (6.77–7.75)		1.28 (1.17–1.41) *p* < 0.001
Stroke/Transient ischemic attack Incidence/1000 people per year (CI95%)	310 4.27 (3.81–4.77)	133 6.7 (5.60–7.93)	352 3.60 (3.23–3.99)	125 6.37 (5.31–7.6)	1.18 (1.01–1.38) *p* < 0.030	1.04 (0.82–1.33) *p* = 0.7444
Heart Failure Incidence/1000 people per year (CI95%)	708 9.75 (9.05–10.50)	476 23.94 (21.84–26.2)	882 8.40 (7.83–8.99)	442 22.54 (20.5–24.74)	1.08 (0.98–1.2) *p* = 0.1224	1.06 (0.93–1.20) *p* = 0.3778
Ischemic Heart Disease Incidence/1000 people per year (CI95%)	816 11.24 (10.48–12.04)	235 11.82 (10.36–13.43)	536 5.48 (5.02–5.96)	141 7.2 (6.05–8.48)	2.05 (1.84–2.28) *p* < 0.001	1.64 (1.33–2.02) *p* < 0.001
Peripheral Arteriopathy Incidence/1000 people per year (CI95%)	753 10.37 (9.65–11.14)	216 10.87 (9.46–12.41)	470 4.80 (4.38–5.26)	122 6.22 (5.17–7.43)	2.16 (1.92–2.42) *p* < 0.001	1.74 (1.4–2.18) *p* < 0.001
Cognitive Impairment Incidence/1000 people per year (CI95%)	440 6.06 (5.51–6.66)	136 6.84 (5.74–8.09)	904 9.24 (8.64–9.86)	154 7.85 (6.66–9.20)	0.65 (0.58–0.73) *p* < 0.001	0.87 (0.7–1.1) *p* = 0.2650
Chronic Kidney Disease Incidence/1000 people per year (CI95%)	1042 14.36 (13.50–15.26)	359 18.06 (16.24–20.03)	1406 14.37 (13.62–15.14)	553 28.20 (25.9–30.65)	0.99 (0.90–1.08) *p* = 0.9965	0.64 (0.56–0.73) *p* < 0.001
Death all-causes Incidence/1000 people per year (CI95%)	1832 25.24 (24.10–26.46)	585 29.43 (27.09–31.91)	2865 29.27 (28.21–30.37)	563 28.71 (26.4–31.18)	0.86 (0.81–0.91) *p* < 0.001	1.02 (0.91–1.15) *p* = 0.6979

## Data Availability

The data supporting the findings of this study are not currently publicly available but can be requested from the authors upon reasonable request. These data will be available through an institutional repository following the public defense of the corresponding PhD thesis.

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
