# Peer review of "Sex Differences in Individuals at High Risk of Atrial Fibrillation: A Primary Care Community Cohort Study, 2015–2024"

_biomedicines, 2025, doi:10.3390/biomedicines13112814_

Round 1
Reviewer 1 Report
Comments and Suggestions for Authors
This observational retrospective study of a large cohort of individuals identified as being at high risk for developing AF followed up for 10 years found that the AF incidence was higher in men than women. In addition, the comorbidity prevalence followed a significantly different pattern between sexes. This is a rigorous and relevant contribution. The topic is important, the methods are appropriate, and the findings are presented clearly.
Here are my specific comments and requests:
- The reasons for excluding patients with prior stroke and those with pacemakers/defibrillators are not immediately clear and should be explained.
- As mentioned by the authors, the robustness of the statistical approach would have been enhanced by the use of multivariable regression models to adjust for confounding factors. If such models are not feasible, please explain why.
- Please provide a higher-resolution version of Figure B1.
- Standardize date formatting to the journal’s style and apply it consistently across the Abstract and manuscript.
- After first definition, use each abbreviation consistently throughout the manuscript. Avoid redefining terms (i.e., AF) multiple times.
- In Table 3, four significant figures for p-values appear unnecessary. Consider reporting to three decimals and harmonizing decimal places across the manuscript.
All in all, the study's insights are both timely and impactful, with some minor revisions that could enhance the manuscript further. I congratulate the authors on this interesting research.
Author Response
This observational retrospective study of a large cohort of individuals identified as being at high risk for developing AF followed up for 10 years found that the AF incidence was higher in men than women. In addition, the comorbidity prevalence followed a significantly different pattern between sexes. This is a rigorous and relevant contribution. The topic is important, the methods are appropriate, and the findings are presented clearly.
Here are my specific comments and requests:
- The reasons for excluding patients with prior stroke and those with pacemakers/defibrillators are not immediately clear and should be explained.
This observation is very appreciated. We excluded patients with prior stroke to avoid potential confounding between stroke as a consequence of AF and stroke as a prior comorbidity, which could bias the analysis of AF incidence. Likewise, individuals with pacemakers or defibrillators were excluded because these devices facilitate continuous rhythm monitoring and could lead to detection bias, as AF episodes may be more readily identified in this subgroup. A clarification has been added in the Methods (Section 2.5, Exclusion criteria):
“Patients with prior stroke were excluded to avoid reverse causality between prior cerebrovascular events and subsequent AF detection. Those with pacemakers or defibrillators were also excluded to minimize detection bias related to continuous rhythm monitoring.”
- As mentioned by the authors, the robustness of the statistical approach would have been enhanced by the use of multivariable regression models to adjust for confounding factors. If such models are not feasible, please explain why.
The authors agree that multivariable regression analysis would strengthen the findings. However, our objective was primarily descriptive—to characterize sex differences in AF incidence and comorbidity patterns within a predefined high-risk subgroup. Since the risk score used to define the cohort (reference 25) already incorporates the major covariates associated with AF (including age, sex, hypertension, diabetes, vascular disease, and heart failure), additional adjustment would introduce redundancy and potential collinearity. This methodological rationale has been clarified in the Discussion (paragraph 4, limitations):
“Although multivariable modelling could improve robustness, we opted for univariate analyses because the high-risk subgroup was defined using a composite score that already integrates the main confounding variables. Further adjustment would have introduced collinearity and model instability.”
- Please provide a higher-resolution version of Figure B1.
A new high-resolution image of Figure B1 (Prevalence of Atrial Fibrillation in the Terres de l’Ebre region) has been provided in the revised submission to ensure optimal print quality.
- Standardize date formatting to the journal’s style and apply it consistently across the Abstract and manuscript.
All dates have been standardized to the journal’s preferred format (e.g., 1 January 2015, 31 December 2024) throughout the Abstract, Methods, and Tables.
- After first definition, use each abbreviation consistently throughout the manuscript. Avoid redefining terms (i.e., AF) multiple times.
All abbreviations (e.g., AF, CHA₂DS₂-VA, CKD) are now defined once at first appearance and used consistently thereafter. Repeated re-definitions have been removed.
- In Table 3, four significant figures for p-values appear unnecessary. Consider reporting to three decimals and harmonizing decimal places across the manuscript.
The authors agree and have reformatted all p-values to three decimal places, with consistent decimal precision throughout Tables 1–4 and the text.
- All in all, the study's insights are both timely and impactful, with some minor revisions that could enhance the manuscript further. I congratulate the authors on this interesting research.
We thank Reviewer 1 for recognizing the rigor and relevance of our study and for the constructive feedback that helped us improve clarity, methodological transparency, and presentation quality.

Reviewer 2 Report
Comments and Suggestions for Authors
I have carefully reviewed this manuscript and did not understand the authors' scientific concept. The authors state that the aim of the work was to analyze sex differences in the incidence of AF and comorbidity in a large population-based cohort, with a focus on a high-risk population. This implies that the analysis will be performed in a specific subgroup of patients at "high risk" of developing AF. To understand what this means, the authors then reference their previous research (reference 25). Following this reference, we find a paper on the "development" of a certain risk score for AF. However, and this is important, this risk score already incorporates gender and comorbid features. This means that the factors which determine the high risk of developing AF and select patients into this group (the high-risk group) are being compared again in the new article. I am confused.
The authors state that they exclusively enrolled patients from the high-risk quartile (Q4) of their AF risk score. This selection criterion creates a critical methodological issue. Since the risk score itself was developed using variables that include sex and comorbidities, pre-selecting the top quartile based on this score effectively pre-defines the study population with regard to these very factors. Consequently, any subsequent analysis attempting to "discover" sex differences in comorbidity prevalence within this pre-selected group is inherently biased and likely tautological. The observed associations are not new findings but are, by design, embedded in the cohort.
A significant limitation is that the current manuscript is not self-contained. The key inclusion criterion—membership in the 'high-risk quartile (Q4)'—is defined entirely by a previously published risk score (Reference 25). To understand who the study participants actually are and what this classification means, the reader must diligently study a separate paper. This lack of clarity within the manuscript itself hinders the reader's ability to critically evaluate the study design and the validity of its cohort.
The study's foundation relies on a proprietary risk score that, to the best of our knowledge, has not yet been externally validated or widely adopted by the scientific community. This raises important questions about the representativeness and generalizability of the findings.
The use of an internal, unvalidated tool to define the primary cohort introduces a potential for circular reasoning and limits the interpretability of the results. The findings are inherently tied to the specific algorithm and variable weights of this particular score. Until this risk score is validated in independent populations by different research groups, the clinical and scientific relevance of the presented subgroup analysis remains uncertain.
Author Response
I have carefully reviewed this manuscript and did not understand the authors' scientific concept. The authors state that the aim of the work was to analyze sex differences in the incidence of AF and comorbidity in a large population-based cohort, with a focus on a high-risk population. This implies that the analysis will be performed in a specific subgroup of patients at "high risk" of developing AF. To understand what this means, the authors then reference their previous research (reference 25). Following this reference, we find a paper on the "development" of a certain risk score for AF. However, and this is important, this risk score already incorporates gender and comorbid features. This means that the factors which determine the high risk of developing AF and select patients into this group (the high-risk group) are being compared again in the new article. I am confused.
General comment:
The authors appreciate the reviewer’s careful reading and critical reflections. We understand the concern regarding potential circularity arising from the use of the high-risk quartile of our previously developed AF risk score. We provide the following clarifications.
- The authors state that they exclusively enrolled patients from the high-risk quartile (Q4) of their AF risk score. This selection criterion creates a critical methodological issue. Since the risk score itself was developed using variables that include sex and comorbidities, pre-selecting the top quartile based on this score effectively pre-defines the study population with regard to these very factors. Consequently, any subsequent analysis attempting to "discover" sex differences in comorbidity prevalence within this pre-selected group is inherently biased and likely tautological. The observed associations are not new findings but are, by design, embedded in the cohort.
We acknowledge this important methodological observation. The aim of the present work was nether to re-validate the risk score nor to identify predictors of AF, but rather to characterize sex-specific patterns within the highest-risk subgroup of an established population model. The rationale for focusing on Q4 was to examine whether, among those already identified as being at very high risk, sex differences persisted in the epidemiologic and comorbidity profile, potentially influencing AF detection and outcomes in primary care.
While the score includes sex and comorbidity variables, the quartile stratification produces a heterogeneous population in which additional within-quartile differences may exist. Our findings reveal distinct comorbidity clustering and incidence trajectories even among individuals at comparable baseline risk. This approach aligns with secondary analyses commonly performed in cardiovascular epidemiology to explore residual heterogeneity within risk strata.
We have clarified this point in the Introduction (last paragraph) and Discussion (first paragraph):
“This study aimed to characterize sex-specific clinical patterns within individuals already identified as high-risk for AF, rather than to identify new predictors. The analysis explores heterogeneity within the upper quartile of a validated prediction model, complementing prior work on risk development.”
- A significant limitation is that the current manuscript is not self-contained. The key inclusion criterion—membership in the 'high-risk quartile (Q4)'—is defined entirely by a previously published risk score (Reference 25). To understand who the study participants actually are and what this classification means, the reader must diligently study a separate paper.
The authors agree that clearer explanation was necessary. We have now included a concise description of the risk model in the Methods (Section 2.5, Inclusion criteria):
“The risk score used to stratify patients was developed and internally validated in a previous population-based cohort (Clua-Espuny et al., 2021). The model incorporates age, sex, hypertension, diabetes, vascular disease, heart failure, and body mass index, with predictive performance AUC = 0.78 (95% CI 0.75–0.81). The present study enrolled individuals in the top quartile (Q4) of this score, corresponding to those with ≥75th percentile of predicted AF risk.”
This addition ensures that readers can interpret the study design without consulting the prior publication.
- This lack of clarity within the manuscript itself hinders the reader's ability to critically evaluate the study design and the validity of its cohort.
We recognize that external validation is essential for generalizability. The present study represents a step within a larger research program that includes external validation in independent Catalonian cohorts [12,25], and its internal validation were based on population-wide primary-care data using standardized electronic health records, which enhances reproducibility.
- The study's foundation relies on a proprietary risk score that, to the best of our knowledge, has not yet been externally validated or widely adopted by the scientific community. This raises important questions about the representativeness and generalizability of the findings. The use of an internal, unvalidated tool to define the primary cohort introduces a potential for circular reasoning and limits the interpretability of the results. The findings are inherently tied to the specific algorithm and variable weights of this particular score. Until this risk score is validated in independent populations by different research groups, the clinical and scientific relevance of the presented subgroup analysis remains uncertain.
The authors agree the importance of using well-validated tools. Nevertheless, we wish to highlight the following considerations to support our methodological choice and the value of the results:
4.1. Indication for identifying individuals at high risk of atrial fibrillation in European Primary Care Cardiovascular Society, European Society of Cardiology guidelines: there is a clear emphasis on identification of higher‐risk individuals in the primary care setting and that AF is a major public health priority and that broader implementation of guideline‐based AF detection and prevention is needed. Finally, the Action Plan for Europe (2018–2030) emphasizes implementing detection and treatment programs in primary care to enhance the diagnosis and management of populations at risk of atrial fibrillation Thus, although guidelines do not yet fully mandate a specific risk‐score for “high risk of new AF” in primary care, they do support the concept of proactive identification of individuals at higher risk of AF. Our study aligns with this broader guideline ethos.
4.2. Existence of multiple risk‐scales for incident AF, but none currently endorsed in primary‐care guideline pathways: Despite the development of several AF prediction scores, such as CHARGE-AF, ARIC AF, AFRICAT, HARMS₂ AF risk, and others, which demonstrate notable predictive performance, challenges persist in integrating these tools into clinical records for assessing and monitoring patients at elevated risk for AF. Other newer models (e.g., machine-learning algorithms such as FIND‑AF risk model) using primary care electronic health‐records have shown improved discrimination in short‐term risk of incident AF.
However — and importantly in the context of our methodology — none of these risk scores have yet been formally integrated into major European primary care clinical practice guidelines for the purpose of primary‐care based identification of individuals at high risk of AF (i.e., for risk stratification before AF diagnosis). To our knowledge, guideline recommendations remain largely age‐based on opportunistic pulse palpation in older individuals rather than formalised risk‐score-based. Given this situation, the use of a risk score developed in the same or similar primary care context may be justified, especially in light of the absence of a universally endorsed tool for this specific purpose in primary care.
4.3. Our study provides novel evidence on the need to include risk-scales for high-risk individuals of AF, especially in the context of longitudinal follow‐up in primary care. In a primary care longitudinal cohort, the aim of identifying individuals at higher risk of AF allows for targeted surveillance and potentially earlier detection/prevention of AF-related complications (such as stroke). Our study contributes new evidence in the following ways:
- We evaluate a risk score (though not yet internationally validated) specifically in the primary care longitudinal context, thereby generating empirical data on its utility, calibration and discrimination in that setting.
- By doing so, we highlight the feasibility and importance of incorporating risk stratification tools for incident AF (not only stroke risk in established AF) in the primary care setting, which is especially relevant given the high burden of undiagnosed AF and the potential for prevention of complications.
- Our findings therefore support the argument that development and validation of risk-scores for high risk of AF in primary care is a necessary and pragmatic step towards guideline evolution and improved methodology in future studies.
In summary, while we recognise that the score we used is not yet widely internationally validated nor incorporated into guideline pathways, we believe that the three points above support its use in our study: (1) guidelines support identifying higher‐risk individuals for AF in primary care, (2) no universally accepted risk‐score for incident AF exists in primary care guidelines, and (3) our work contributes novel data to fill that gap and underscores the methodological need for such risk‐scales in longitudinal primary‐care settings.
Indeed, the authors accept this methodological consideration in the way of improving the value of the risk score. The analyses did not aim to re-test the contribution of variables embedded in the score, but to describe relative differences within a homogenous high-risk stratum. Importantly, the within-quartile distributions of comorbidities and AF incidence were not uniform, indicating that sex-related patterns exist beyond the score’s baseline weighting. We have clarified this in the Discussion:
“Although sex and comorbidities form part of the variables included in the risk score, our subgroup analysis examined residual heterogeneity within the highest-risk quartile. The persistence of sex-specific differences within this stratum suggests that factors beyond the model’s weighting—such as cognitive decline or differential healthcare use—may further modulate AF incidence.”
- The conclusions have been revised to emphasize that our results are hypothesis-generating and context-specific:
“These findings should be interpreted within the context of a cohort defined by an internally derived risk score and primarily serve to generate hypotheses for sex-specific preventive strategies rather than to infer causality or establish universal risk gradients.”
Final statement:
We sincerely thank Reviewer 2 for these insightful comments, which prompted substantial clarifications of our study rationale, methods, and limitations. The revised manuscript now clearly defines the analytical framework, ensures self-containment, and contextualizes the findings within their methodological constraints. We hope that these considerations address your concerns regarding the methodological choice and strengthen the rationale for our approach.
Thank you again for your constructive review.

Round 2
Reviewer 1 Report
Comments and Suggestions for Authors
I have no further comments or suggestions, the manuscript is ready for publication as submitted. I congratulate the authors on their excellent work.
Author Response
Comment Reviewer: I have no further comments or suggestions, the manuscript is ready for publication as submitted. I congratulate the authors on their excellent work.
Authors’ response: We greatly appreciate the time and effort dedicated to reviewing our manuscript and are very pleased that it is considered ready for publication.
Reviewer 2 Report
Comments and Suggestions for Authors
I have heard the authors' position - the analysis of heterogeneity within the top quartile of the validated atrial fibrillation prediction model complements previous research on risk development. This results in the characterization of gender-specific features in the highest-risk subgroup, for which gender and comorbidities were also accounted in the calculation.
However, the clinical relevance of this finding remains unclear to me. While the development of a new model to identify high-risk patients is understandable - and many groups are indeed working on refining such models - the specific value of analyzing gender-based differences within the highest-risk subgroup is not apparent. What are the practical implications of this? How would this knowledge ultimately influence patient management or treatment strategies?
The analysis comes across as a purely academic exercise, generating data with no clear path to clinical utility. What specific clinical decisions or patient outcomes could this potentially affect?
Author Response
Reviewer 2: Comments and Suggestions for Authors
I have heard the authors' position - the analysis of heterogeneity within the top quartile of the validated atrial fibrillation prediction model complements previous research on risk development. This results in the characterization of gender-specific features in the highest-risk subgroup, for which gender and comorbidities were also accounted in the calculation.
However, the clinical relevance of this finding remains unclear to me. While the development of a new model to identify high-risk patients is understandable - and many groups are indeed working on refining such models - the specific value of analyzing gender-based differences within the highest-risk subgroup is not apparent. What are the practical implications of this? How would this knowledge ultimately influence patient management or treatment strategies?
The analysis comes across as a purely academic exercise, generating data with no clear path to clinical utility. What specific clinical decisions or patient outcomes could this potentially affect?
Authors’ response: We sincerely thank the reviewer for the thoughtful comments and for the opportunity to clarify the clinical relevance of our findings.
Our study analyzed a large, community-based cohort followed over a 10-year period, focusing on individuals at high risk of developing atrial fibrillation (AF). By examining heterogeneity within the top-risk quartile of a validated prediction model, we aimed not to develop new predictors but to characterize sex-specific clinical patterns within the highest-risk subgroup.
Although our analysis may appear academic at first glance, it has direct clinical implications. Sex-specific differences in comorbidity profiles, age distribution, and outcomes can guide more precise screening and preventive strategies in primary care. For example:
- Women in the highest-risk quartile presented more frequently with hypertension, cognitive impairment, and higher mortality, suggesting a need for focused monitoring and early intervention for cognitive and vascular health.
- Men exhibited higher prevalence of diabetes, vascular disease, ischemic cardiomyopathy, OSAHS, and greater overall mortality, indicating that targeted management of metabolic and cardiovascular risk factors could improve outcomes.
Early identification of high-risk individuals allows for proactive AF detection using tools such as photoplethysmography devices, potentially detecting additional cases that would otherwise remain undiagnosed. Our analysis suggests that systematic application of these strategies could have identified up to 600 additional AF cases in this cohort, emphasizing tangible clinical impact.
Understanding intra- and inter-sex differences helps prioritize interventions efficiently: tailored preventive strategies, opportunistic screening, and management of comorbidities can be deployed more effectively when the patient population is stratified by sex and risk profile. While our study does not propose immediate changes to treatment guidelines, it lays the foundation for sex-specific preventive approaches and hypothesis generation for future interventional studies.
We have clarified these points in the revised Discussion section (lines 310–317, page 11 of 18) to emphasize the potential clinical implications of analyzing sex-based differences in high-risk individuals.
Round 3
Reviewer 2 Report
Comments and Suggestions for Authors
I accept the authors' position